# Mechanical Insufflation-Exsufflation: Considerations for Improving Clinical Practice

**DOI:** 10.3390/jcm12072626

**Published:** 2023-03-31

**Authors:** Michelle Chatwin, Ruth Helen Wakeman

**Affiliations:** 1NMCCC, The National Hospital for Neurology and Neurosurgery, University College London Hospitals Foundation Trust, London WC1N 3BG, UK; 2Clinical and Academic Department of Sleep and Breathing, Royal Brompton Hospital, Part of Guys and St Thomas’ NHS Foundation Trust, London SW3 6NP, UK; 3Department of Paediatrics, Royal Brompton Hospital, Part of Guys and St Thomas’ NHS Foundation Trust, London SW3 6NP, UK

**Keywords:** cough assist, cough, neuromuscular disease, cough peak flow, airway clearance techniques, MI-E, bulbar insufficiency

## Abstract

The provision of mechanical insufflation-exsufflation (MI-E) devices to enhance cough efficacy is increasing. Typically, MI-E devices are used to augment cough in patients with neuromuscular disorders but also in patients who are weak in an acute care setting. Despite a growing evidence base for the use of these devices, there are barriers to the provision of MI-E, including clinician lack of knowledge and confidence. Enhancing clinician education and confidence is key. Individualized or protocolized approaches can be used to initiate MI-E. Evaluation of MI-E efficacy is critical. One method to evaluate effectiveness of MI-E is the MI-E-assisted cough peak flow (CPF). However, this should always be considered alongside other factors discussed in this review. The purpose of this review is to increase the theoretical understanding of the provision and evaluation of MI-E and provide insight into how this knowledge can be applied into clinical practice. Approaches to initiation and titration can be selected based on the clinical situation, patient diagnosis (including and beyond neuromuscular disorders), and clinician’s confidence.

## 1. Introduction

Respiratory secretions are cleared via the mucociliary escalator. Breathing naturally enhances secretion movement; as when we breathe in, the flexible airways expand and when we breathe out, they narrow. This creates an expiratory airflow bias that assists in moving the secretions in a cephalad direction [1,2,3]. Secretion retention can occur because of an increased production of secretions due to the physiological consequences of lung disease or due to a weak cough. In patients with a weak cough, retention of secretions is a major cause of mortality and morbidity. Patients with a weak cough are typically patients with neuromuscular disease. Airway clearance techniques (ACTs) can eliminate secretions and improve survival in this patient group. Patients who have respiratory muscle weakness and retain secretions should be assessed by a specialist respiratory physiotherapist or respiratory therapist depending on the local provision of care. For this review, they will be referred to as the respiratory physiotherapist (RP). During the assessment, the RP will evaluate where secretions are situated and apply the most suitable ACT. Lannefors et al. [4] described a four-step process to assist in the decision making for airway clearance. Chatwin et al. [5] highlighted that stages 1 to 3 are peripheral ACTs (secretion mobilizing), and stage 4 is a proximal ACT (cough augmentation). Prior to initiating proximal ACTs, it is important to assess cough function. This enables the RP to determine the most suitable ACT. This review describes the process of provision of mechanical insufflation-exsufflation devices (MI-E), from cough assessment to critical cough peak flow levels and MI-E initiation. A clear understanding of the theoretical basis for MI-E provision can support clinicians in optimizing its application for airway clearance, including and beyond patients with neuromuscular disorders.

## 2. Cough Assessment

A normal cough consists of four phases as shown in Figure 1. A cough is also audible and the sound that is generated during the cough can be used to judge whether the cough is strong enough to clear secretions [6]. It is always important to ask your patient about their cough and whether they have difficulty clearing secretions. This is because a cough can be effective when well, but due to the decline in respiratory muscle strength as a result of a cold or respiratory tract infection, can become ineffective [7]. Further assessment should occur observing the inspiratory and expiratory muscle activity and glottic function along with a measurement of the unassisted cough peak flow (CPF) (see Figure 2). The easiest way to measure an unassisted CPF is with a pediatric peak expiratory flow meter attached to an anesthetic mask [5,8]. A pediatric flow meter is more accurate at low flows than an adult one. The patient is asked to take a deep breath in, and the mask is placed firmly over their nose and mouth, then they are asked to cough hard. This is repeated and the highest value is taken to be their CPF [5,8].

## 3. Cough Peak Flow Values

There are reference values based on centiles for children [9]. In adult patients, a normal cough strength exceeds 360 L/min [10]. An assisted CPF of 160 L/min has been deemed to be the level required to prevent retention of secretions [11]. This level is based on a study looking at patients who successfully cleared their secretions after removal of their tracheostomy. In patients where an assisted CPF was not > 160 L/min, they required re-insertion of their tracheostomy to clear their secretions [11]. An assisted CPF of < 270 L/min in those over the age of 12 years old is a warning sign that the cough strength could rapidly deteriorate to a critical level of 160 L/min [12,13]. Unassisted peak cough flows are a useful tool to determine what proximal airway clearance techniques (ACT) or cough augmentation techniques, as they are also known, should be used and when [8].

## 4. Proximal Airway Clearance Techniques

Toussaint et al. [8] highlighted six cough augmentation techniques and the CPF ranges that these treatment options could be targeted based on the evidence at the time. When the CPF is <180 L/min, then MI-E is first choice, and in the weaker patients or to enhance efficacy, these techniques should be combined with a manual assisted cough. It is important not to use MI-E in all patients with high CPF (>5 L/sec) [14]. This is because the expiratory flow produced by the patient’s effort and manual assisted cough transiently exceed the vacuum capacity of the MI-E device, which therefore becomes a transient load against the CPF [14].

## 5. Mechanical Insufflation-Exsufflation Devices

MI-E should be targeted at weaker patients (CPF < 180 L/min) or after other techniques have failed [5,8]. A MI-E device delivers positive pressure, followed by a rapid switch to negative pressure. This simulates the flow changes that naturally occur during a cough, increases inspiratory and expiratory volumes, and therefore increases expiratory flows. This moves secretions up towards the mouth until they are high enough to come into the mouth or be suctioned from the mouth or via the nasopharyngeal passage. There are various devices on the market, and they all work in much the same way. The device is connected via a 22 mm tube to either an oronasal mask, mouthpiece, or catheter mount. Therefore, the device can be used in patients with or without an artificial airway. Patients are asked to let the device fill up their lungs and then cough if needed when it sucks out. Contra-indications and precautions are shown in Box 1. Initial settings should be reviewed and adapted for continual effectiveness and patients may need higher settings when unwell due to an increase in their respiratory muscle weakness [7].

Box 1Contraindications and precautions to consider when commencing MI-E therapy.
Contraindications


History of bullous emphysema, surgical emphysema, pneumothorax—undrained, known susceptibility to pneumothorax, care with patients who have had a pneumothorax, Pneumo-mediastinum, recent barotrauma, patients known to have cardiac instability should be monitored closely for pulse and oxygen saturation when using the MI-E device, tracheoesophageal fistula


Precautions


History of pneumothorax, recent lobectomy/pneumonectomy, cardiovascular insufficiency, acute abdominal distention, poor patient cooperation, pulmonary air leak



These contraindications and precautions are listed in the instructions for use of the Clearway 2 [15] (Breas Medical, Mölnlycke, Sweden) and E70 [16] (Philips Respironics, Murrysville, PA, USA) and by Swingwood and coworkers [17].

## 6. Patient Populations and Clinical Benefits of MI-E Devices

Table 1 presents signposts to some of the key physiological, clinical, and quality of life benefits of MI-E. MI-E has been shown to increase CPF in patients with neuromuscular disorders (NMD) [18,19,20,21], shorten airway clearance treatment times [22,23], and decrease treatment failure [24,25] in acute care. Studies in acute care (outside the NMD’s population) have also reported increased sputum clearance [26,27], leading to improvements in lung compliance [26,27]. MI-E seems to decrease respiratory events and time in hospital in patients with NMDs [28]. In conjunction with non-invasive ventilation (NIV), it has increased survival in amyotrophic lateral sclerosis (ALS) patients [29]. MI-E may be beneficial to patients with cerebral palsy, but further work is warranted [23]. More recently, based on clinical experience, and where indicated, MI-E has been recommended in the BTS clinical statement on the prevention and treatment of community acquired pneumonia in individuals with learning disability [30].

## 7. Barriers to the Use of MI-E

Despite a substantial evidence base for the use of MI-E, there are still barriers to its use. Rose and co-workers [47] surveyed UK and Canadian physio/respiratory therapists for their views around the use of MI-E in the long-term setting. The main barrier reported was insufficient funding for equipment, followed by insufficient knowledge and familiarity of the devices by teams. Access to equipment in a timely manner and inability to provide support and follow-up were also cited. In the same year of publication of the survey, the ENMC International workshop on airway clearance techniques in NMDs was published [8], along with a state-of-the-art paper [5]. The purpose of these publications was to provide evidence-based, where possible, or consensus recommendations for airway clearance techniques in NMDs, with the aim of increasing knowledge and improving patient care in both acute and long-term care. MI-E use was recommended in acute care in these papers for patients with NMDs [5,8]. We know that patients in the ICU are weak and have an impaired cough due to an artificial airway. Unmanageable secretion load has been reported in 89% of patients requiring re-intubation, in comparison to only 39% of those who were successfully extubated [48]. In acute care, the provision of MI-E in the intensive care (ICU) environment in the Netherlands was only 22% [49]. Swingwood and co-workers [50] also surveyed UK use of MI-E in the ICU and similar barriers as Rose and co-workers [47]. These barriers included no experience, insufficient evidence, lack of confidence, and lack of clinical need. Interestingly, they reported that clinicians were more likely to use MI-E in the patient without an artificial airway [47]. Swingwood and colleagues [17] also reviewed all the papers where MI-E was used in the ICU. From these papers, perceived lack of skills and knowledge and resources were also identified as a barrier; however, they also identified enabling features to utilizing MI-E, in that if they had a positive clinical response to MI-E, then they would be more likely to use it. A positive team culture was also an enabler to adoption of MI-E within clinical practice.

## 8. Effective MI-E Pressures

Bach, in 1994 [41] and 2002 [51], highlighted that MI-E devices should be used with pressures around +40 and −40 cm H_2_O. In the same year, Gomez-Merino and co-workers [52] carried out a bench study evaluating the effect of different MI-E pressures and times on inspiratory and expiratory volumes and flows. To generate high expiratory flows, higher pressures were required (+40 to −40 cm H_2_O) along with higher inspiratory to expiratory times (3 s to 2 s). These settings generated an expiratory flow of 4.09 L/sec or 245 L/min [52]. Hyun and co-workers [32] evaluated CPFs generated with and without an artificial airway, again higher pressures were required to generate the greatest expiratory flows. However, higher flows were seen for the same pressure when MI-E was delivered via the upper airway rather than the artificial airway. The likely explanation for this is that the glottis can close when the artificial airway is removed. This allows a build-up of intrathoracic pressure which is not possible in the presence of an artificial airway. The authors also concluded that there was no haemodynamic instability with pressures up to +50 and −50 cm H_2_O and there were no complications of pneumothorax or pneumomediastinum. In a pig model, Marti and co-workers [34] evaluated the movement of metal (tantalum) disks. They found that settings of +40 to −70 cm H_2_O increased artificial mucus velocity almost 5-fold. They reported a transient significant increase in inspiratory pulmonary pressure at setting ≥ 50 cm H_2_O. Despite this, no adverse events were reported. So, taking this work into account, in weaker patients, or those who do not cough at the point of exsufflation, or those with an artificial airway, higher pressures to generate effective assisted CPF are required.

There are various ways to commence MI-E, utilizing pressures of +40 and −40 cm H_2_O and this has been reported to be effective in studies [18,25,27,53,54,55,56] and is utilized more in North America. There is also a more individualized approach [18,19,23,37,43,57,58,59], which has predominately been highlighted in Europe. However, from personal experiences when teaching about the practicalities of MI-E initiation, individuals who are less confident or new to initiating MI-E may find individualization of settings daunting and lack the confidence to do this. One option to build up confidence is to utilize protocols for the initiation of MI-E, an example of one such approach is shown in Figure 3.

One alternative approach, which has been reported more in Europe, is to individualize settings for the patient. However, as previously stated, for RPs new to initiating MI-E, this may cause anxieties as there are more steps to the initiation process. An alternative option is for the clinician to gain confidence with a more protocolized approach and once they have been successful, to individualize. Reasons for suggesting a more individualized approach include that bench [60] and clinical case series [57,59] have highlighted a greater negative to positive pressure that has been clinically effective. When it comes to the insufflation pressure, Mellis and Goebel [61] showed that it is not necessarily the deepest breath in that produces the highest CPF. Therefore, the insufflation pressure should be titrated to patient comfort. Chatwin and Simonds [57] produced an algorithm on how to set up MI-E based on their clinical practice. Much of this algorithm continues to be appropriate for the adult population today. However, a longer inspiratory than expiratory time is likely to enhance inspiratory volumes and expiratory flows [52]. Figure 4 is an updated version of their original algorithm incorporating the changes in practice and considering more appropriate inspiratory and expiratory times. Once confidence has been achieved with individualizing settings by increasing the expiratory pressure, clinicians can review other settings if they suspect upper airway closure that has been reported via laryngoscopy [43,44] or flow volume curves [62]. Further individualization can be important in specific disease groups. Andersen and co-workers [43,44] suggested in ALS patients who experience upper airway closure, to adjust the following parameters with the aim of preventing airway closure: ensuring triggering the inspiratory pressure, decreasing the inspiratory flow, decreasing the inspiratory pressure, and increasing the inspiratory time. The rationale for these recommendations is that patient-triggered insufflations lead to less laryngeal adduction and in other patient groups, can enhance co-ordination. Swallowing reflexes can be triggered by the insufflation pressure. Lower insufflation pressures can be help this along with decreasing the inspiratory flow [44]. However, decreasing the inspiratory pressure will decrease the pre-cough inspiratory volume to the patient. To accommodate this, the inspiratory time can be lengthened, which will increase the inspiratory volume. A reduction in the inspiratory flow has also been shown by Volpe and co-workers [63] in a bench model to increase the expiratory flow bias and enhance secretion movement, when using the same pressures. Therefore, reducing the inspiratory flow can be useful in patients who find it difficult to tolerate high pressures but have a large volume of secretions to clear.

Although laryngoscopy is the gold standard to review what is happening to the upper airway during MI-E treatment, not all clinicians have access to laryngoscopy to assess the efficacy of MI-E in a subset of patients who experience ineffective treatment. In the original algorithm by Chatwin and Simonds [57], it was highlighted to look at the CPF produced with MI-E. However, based on recent publications [62,64], this may not be a good outcome measure. This is because there will be an assisted expiratory flow during exsufflation regardless of whether the upper airway is open or closed. Lamolda and co-workers [58] described what happens during a MI-E-assisted cough. The initial flow that is sucked out is the compressible volume (the air that is in the upper airway and circuit and that this usually occurs in the first 100 ms (see Figure 5)). So, if the airway is closed on either insufflation, exsufflation, or both, then the compressible volume peak can often be mistaken for the real CPF. Reviewing of flow and pressure can help a clinician determine whether the airway is open or closed and whether obstruction occurs during insufflation, exsufflation, or both [58,62,64]. Identification of this in the flow traces requires education and training. Education for clinicians does not always focus on this potential variation between the CPF seen on devices and the true CPF, see Figure 6. Perhaps one answer is that devices should incorporate algorithms to avoid a false CPF reading (removing the compressible volume). If a clinician is unable to download or feel confident in reviewing the traces, an alternative method is to auscultate the trachea. Auscultation is widely used in respiratory therapy/physiotherapy, and therefore can be easily adopted into evaluation of MI-E. When auscultating over the trachea air can usually be heard moving in and out. If airflow is not heard whilst the insufflation is delivered, there is obstruction on insufflation. Sancho and co-workers [62] agree with Andersen and co-workers [43] to individualize the patient’s settings further by lowering the inspiratory pressure and flow. However, if air is not heard when negative pressure is applied there is obstruction with exsufflation. In this situation the exsufflation pressure should be decreased. If obstruction occurs on both, then the inspiratory and expiratory pressures and the inspiratory flow should be adjusted. Further attention may be needed with regards to the inspiratory and expiratory times to further decrease residual airway obstruction [62]. The upper airway is more likely to be closed on insufflation in predominately the upper motor neuron at the bulbar level and if it is closed on exsufflation, it is predominately lower motor neuron patients at the bulbar level [62].

## 9. MI-E with the Addition of Oscillations

Further individualization of settings can occur when adding oscillations on either insufflation, exsufflation, or both. It is thought that the oscillations as highlighted in high frequency chest wall oscillation (HFCWO) have the potential to decrease sputum viscosity [65]. Typically, peripheral ACTs take 20–30 min to mobilize secretions. Manufactures may highlight MI-E with the addition of oscillations as being a potential substitute for using HFCWO. Sancho and co-workers [66,67,68] report their work looking at the effect of the addition of oscillations on MI-E. In their studies, they used an insufflation pressure of around +40 cm H_2_O and an exsufflation pressure of −40 cm H_2_O, insufflation time 2 s and exsufflation time 3 s. Oscillations were applied with a frequency of 15 Hz with an amplitude of 10 cm H_2_O, usually for two sessions of 6–8 cycles applied each day and additionally when necessary. They found that MI-E-assisted CPF did not increase with MI-E with oscillations in stable ALS patients [68]. Long-term use (1 year) in non-invasively managed and tracheostomized ALS patients, who use MI-E compared to MI-E with oscillations, showed no benefit. They found no decrease in hospitalizations or increase in survival in either group [66,67], and MI-E with oscillations did not reduce the need to perform invasive procedures for mucus removal or decrease the risk of respiratory tract infections nor improve 1-year survival compared to MI-E alone [66,67]. It would make sense that MI-E with the addition of oscillations may help to move secretions. However, the studies by Sancho and coworkers only delivered oscillations for a very short period [66,67,68]. This is because delivering high pressures (those needed to enhance cough) cannot be tolerated for a long time as they will cause hyperventilation. Therefore, based on the current evidence base, there is no indication to use oscillations with MI-E with a protocolized or individualized approach.

If the aim of MI-E with the addition of oscillations is to mobilize secretions acting as a peripheral ACT, then settings need to be modified and individualized for this by the RP. This would involve having lower insufflation and exsufflation pressures but high enough to maintain adequate ventilation but not too high to cause the patient to experience hyperventilation if the device is used for at least 5 min at a time for a total of 20 to 30 min. Decreasing the inspiratory flow may also help [63]. It is important not to recommend the use of MI-E with oscillations in all patients or when first initiating MI-E therapy unless there is clear clinical indication. The use of MI-E with oscillations may be a strategy that is tried by a RP who has extensive experience with the device and is confident in modifying settings as a peripheral airway clearance strategy.

## 10. Adherence

Clinicians may be concerned that patients are not using devices, and as a result, may believe/extrapolate that the settings are incorrect or ineffective. Unlike NIV, there is no defined amount of time that patients should be using MI-E. Bach and co-workers [12,69] used a protocol to manage their patients. The protocol stated they should use MI-E in the presence of secretions and when their oxygen saturations were below 95% with no entrained oxygen. Unfortunately, there was no documented usage of MI-E in these studies. Chatwin and Simonds [57] reviewed adherence to MI-E treatment in their cohort of patients. Despite recommending patients use the device at least once a day, they found MI-E was used differently depending on the secretion burden. Sixty four percent of patients with tracheostomies used MI-E daily versus 31% who did not have a tracheostomy; only SMA type I used the device daily. Mitropoulou et al.’s [70] more recent work also reported that usage of MI-E is related to secretion burden rather than CPF values. Clinicians should make recommendations at initiation around daily MI-E use to build up confidence around the device. However, when reviewing treatment, it is more likely patients will use the device depending on their secretion burden.

## 11. Conclusions

Prior to the provision of proximal airway clearance techniques, a formal cough assessment should take place to assess cough efficacy and strength. MI-E devices should be used in patients who have weak coughs < 160 L/min. MI-E settings of at least +40 to −40 cm H_2_O have been shown to create high enough expiratory flows to provide an effective cough. For clinicians who are less confident in initiating MI-E, a protocolized approach to setting up a device, and building up to pressures of +40 to −40 cm H_2_O where tolerated, will be effective at clearing secretions. When clinicians are more confident in initiating and evaluating MI-E, a more individualized titration can enhance cough efficacy further, especially in ALS patients. MI-E settings should be regularly reviewed, and settings increased when patients are unwell. Reviewing MI-E-assisted CPF should not be used in isolation, as an expiratory flow reading will be recorded whether the airway is open or closed. It would seem MI-E usage is related to secretion burden and devices should not be removed if patients only use them when there are secretions present.

## Figures and Tables

**Figure 1 jcm-12-02626-f001:**
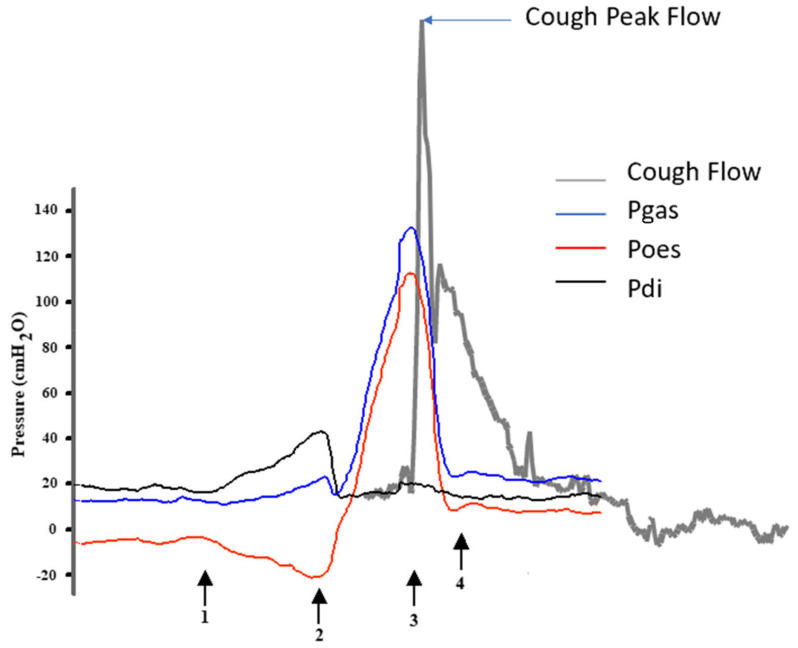
Shows the components that occur within a normal cough. Blue represents the gastric pressure, red is the esophageal pressure, and black is the transdiaphragmatic pressure. Superimposed on this in grey are the flow changes that are seen during a cough. Arrow 1 indicates the start of the deep inspiration. Arrow 2 is the forced expiration against a closed glottis. Arrow 3 is the sudden opening of the glottis and Arrow 4, compression of the intra-thoracic airways. Cough peak flow (CPF) is measured as the peak expiratory flow that occurs during a cough. For anyone over the age of 12 years old, a CPF that exceeds 360 L/min is deemed to be normal. There are reference values for children [9].

**Figure 2 jcm-12-02626-f002:**
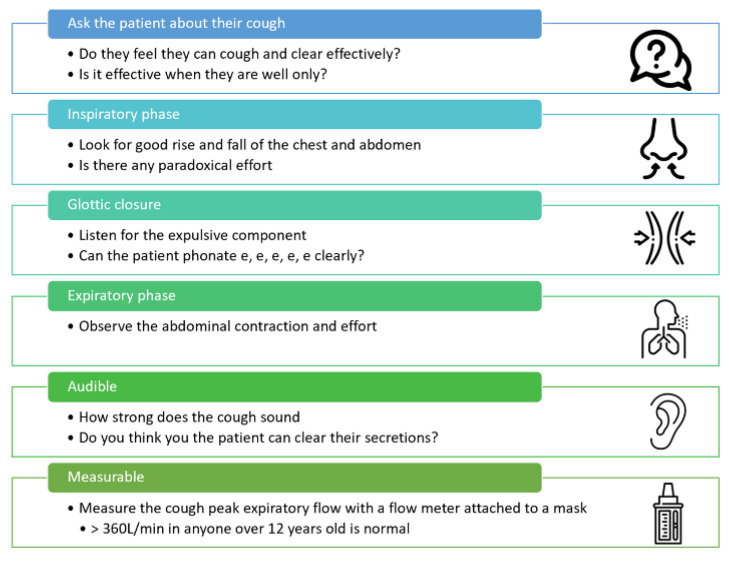
Shows one approach to assessing the efficacy of a patient’s cough.

**Figure 3 jcm-12-02626-f003:**
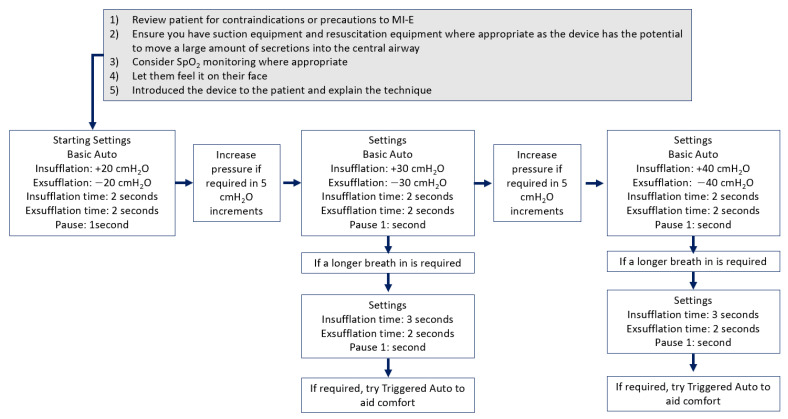
Shows a suggested algorithm for a standard approach to MI-E initiation in adult patients.

**Figure 4 jcm-12-02626-f004:**
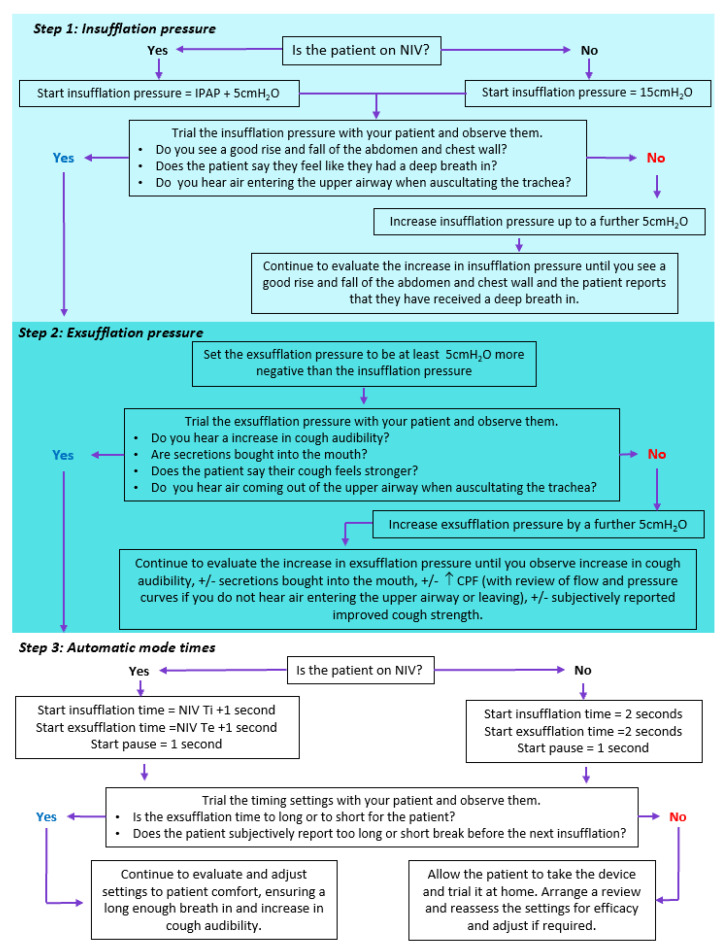
Shows a suggested algorithm for personalized approach to titration of mechanical insufflation-exsufflation (MI-E) devices in adult patients. This algorithm is modified from Chatwin and Simonds [57], taking into account more recent evidence around MI-E times and also MI-E-assisted cough peak flow (CPF). Inspiratory time (Ti), expiratory time (Te).

**Figure 5 jcm-12-02626-f005:**
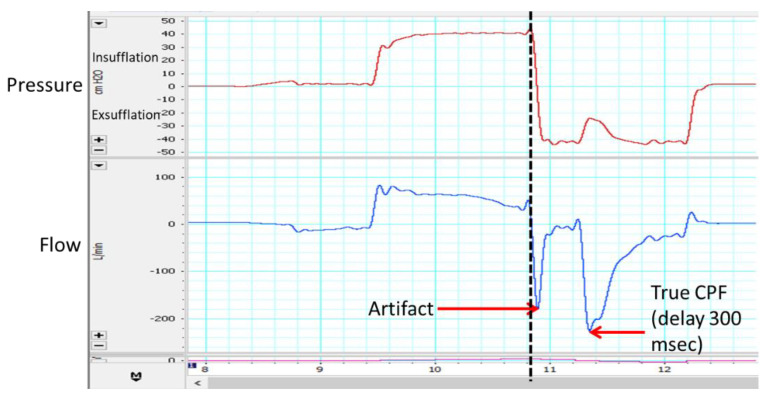
Shows pressure (red) and flow (blue) traces during a cycle of mechanical insufflation exsufflation with a patient coughing. The first negative flow point is artifact as it is the compressible volume. The second peak is the true cough peak flow (CPF). With thanks to Manuel Lujan.

**Figure 6 jcm-12-02626-f006:**
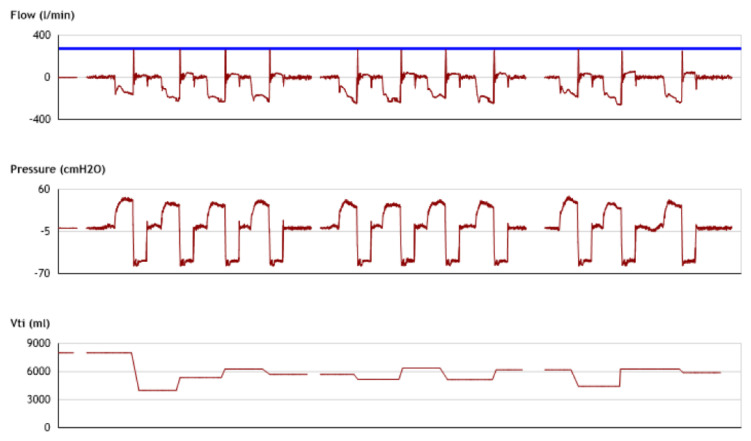
Shows the flow, pressure, and inspiratory tidal volumes (Vti) download for a patient using an E70 (Philips Respironics, Murrysville, PA, USA). The blue line shows the device assisted cough peak flow (CPF) of around 350 L/min. Clinicians would normally be very pleased with this reading as it is at a level that would indicate an effective assisted CPF. However, after the initial CPF, there is flattening of the exsufflation curve, indicating an obstructed airway. With thanks to Tiago Pinto.

**Table 1 jcm-12-02626-t001:** Literature pertaining of physiological, clinical, quality of life benefits and literature complications of MI-E.

Physiological Benefits of MI-E	Reference
Increased cough peak flow	[18,19,20,21,31,32,33]
Increased secretion clearance	[26,34]
Improvements in lung compliance	[26,34]
Short term improvements in forced vital capacity	[35]
**Clinical benefits**	
Decreased hospital admissions, time in hospital or Physicians visits	[36]
Decrease treatment failure	[24,25]
Decreased treatment time	[22,23]
Increased survival (in conjunction with ventilatory support)	[29,37]
Patient preference for MI-E over suction	[38]
**Complications to MI-E**	
Pneumothorax, pneumomediastinum	[39]
Abdominal distention, nausea, bloating	[40]
Bradycardia, tachycardia	[41]
Thoracic wall discomfort	[42]
Upper airway collapse	[43,44,45]
**Quality of life benefits**	
Perceived improvement in overall respiratory health	[46]
Disadvantage was the size of the devices	[46]

## Data Availability

Not applicable.

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
