# Peer review of "Mechanical Insufflation-Exsufflation: Considerations for Improving Clinical Practice"

_jcm, 2023, doi:10.3390/jcm12072626_

Round 1

Reviewer 1 Report

The review article entitled "Mechanical Insufflation-Exsufflation as An Airway Clearance Tool; Protocolized Versus Personalized Approaches," presented by Chatwin M and Wakeman R H, contributes to the "theoretical understanding of the provision and evaluation of MI-E and provide insight into how this knowledge can affect current clinical practice about which approach can be used and in what situation or with which patient group."

First, it is essential to emphasize that this review may clarify the clinical use of MI-E device in "non-conventional" situations (neuromuscular diseases).

Overall, the article was well-written. However, there are some considerations that I recommend minor revision. I've shared my suggestions detailed below:

- Please provide references in Box 1 regarding the contraindications.

- I suggest inserting a table with articles related to MI-E use from at least the last ten years that summarize the Clinical Benefits, indications, barriers, etc.

- I congratulate the authors for the schemes presented in the article.

Author Response

We would like to thank Reviewer 1 for their constructive comments and respond point by point below:

  1. Please provide references in Box 1 regarding the contraindications.

 Many thanks for highlighting this. The references have been added to Box 1.

 2. I suggest inserting a table with articles related to MI-E use from at least the last ten years that summarize the Clinical Benefits, indications, barriers, etc.

 Thank you for this suggestion, we agree that this would be an interesting table. However, to do this correctly it would involve a full systematic review, and this is potential beyond the scope of this manuscript. This would certainly make for an interesting second manuscript.

In addition to these comments we have also clarified within the text that the manuscript includes NMD's but also highlights situations beyond this population.

Thank you for your time in reviewing this manuscript

Reviewer 2 Report

I think this is a good review to increase the theoretical understanding of the provision and evaluation of MI-E and 20 provide insight into how this knowledge can affect current clinical practice. I have some suggestions for the authors.

1. Can the authors give brief titles for every figures?

2. The article is a bit lengthy, and it is suggested to be simplified.

3. Please strengthen the main ideas of your review. 

4. Can the authors deepen the discussion of personalized approaches ?

Author Response

We would like to thank Reviewer 2 for their constructive comments. We have responded point by point below.

I think this is a good review to increase the theoretical understanding of the provision and evaluation of MI-E and 20 provide insight into how this knowledge can affect current clinical practice. I have some suggestions for the authors.

  1. Can the authors give brief titles for every figures?

Unfortunately, the MDPI format does not allow this. We have submitted as per their guidelines

  1. The article is a bit lengthy, and it is suggested to be simplified.

Many thanks for this comment we originally submitted a shorter manuscript and was asked to increase the word count to 4000 words to meet the Journals requirements. That said, we have been through the manuscript and deleted as much content as possible.

  1. Please strengthen the main ideas of your review. 

We hope that we have highlighted the educational aspect of this review to ease use for clinicians who are new to MI-E but also adding aspects to individuals who are proficient in its use.

  1. Can the authors deepen the discussion of personalized approaches?

Many thanks for this comment we have depended the discussion of the personalised approach.

Reviewer 3 Report

Non-invasive methods for eliminating tracheo-bronchial secretions are more desirable to patients than is tracheostomy and gained large popularity in the early twenty-first century. In that period, a substantial number of studies was published evaluating results from the application of non-invasive cough augmentation techniques, particularly Mechanical In-Exsufflation, on patients with respiratory muscle weakness or paralysis.

The goal of this review is to discuss benefits from the application of  MI‑E for airway clearance in patients with ineffective cough and barriers to its use .

Major comments

1.        The title of this review is not well addressed to its content. Definitely, as stated in the conclusions, this review highlights on both assessment of cough efficacy and strategy of MI-E application and not on a comparison between protocolized and personalized approaches;

2.        The  review primarily focuses on findings on the use of MI-E devices which are well-known for about 20 years. Indeed, most references relate to that period.

3.        Content seems to be educational in some parts of the review, that is aimed at educating the audience rather than discussing scientific issues or concerns.

Author Response

We would like to thank Reviewer 3 for their constructive comments. We have responded point by point below.

Major comments

  1. The title of this review is not well addressed to its content. Definitely, as stated in the conclusions, this review highlights on both assessment of cough efficacy and strategy of MI-E application and not on a comparison between protocolized and personalized approaches;

We would like to thank you for bringing this to our attention. We have appended the title and hope this better reflects the topics within the review article.

  1. The review primarily focuses on findings on the use of MI-E devices which are well-known for about 20 years. Indeed, most references relate to that period.

This is correct but we aim to fill in the gap for people who have a low level of confidence when using MI-E and also provide so further information to enhance treatment strategies for those who are more proficient in delivering treatment.

  1. Content seems to be educational in some parts of the review, that is aimed at educating the audience rather than discussing scientific issues or concerns.

Many thanks for this comment you are correct in that this review is a review that helps to improve education of the use of these devices. There is no definitive way this device should be used and in teaching how to use MI-E it has been noted a personalised approach is too difficult for less experienced clinicians. Also, in North America provision of MI-E is by using is with similar insufflation and exsufflation pressure values. We aim to highlight that this effective in a large number of patients as supported by the research and that it does not have to be one method over another. Building up confidence to deliver treatment rather than not to use it due to a lack of confidence.

Round 2

Reviewer 2 Report

I appreciated the authors' response and I have no more questions about this manuscript.

Author Response

We would like to thank you for your time and constructive comments in reviewing our manuscript.

Reviewer 3 Report

The Authors clarified doubts on the educational purpose of their manuscript.

I would suggest the Authors to include a table summarizing major data from published studies on MI-E utilization.

Author Response

We would like to thank reviewer 3 for their valuable comments. We have for educational purposes, inserted table 1 to summarise the key physiological and clinical benefits of MI-E. This is to signpost those who have a lower depth of knowledge to key references they may find useful in reading should they want further information. 
